# Thermogravimetric Analysis (TGA) of Graphene Materials: Effect of Particle Size of Graphene, Graphene Oxide and Graphite on Thermal Parameters

**Farzaneh Farivar** [1,2,†], **Pei Lay Yap** [1,2,†], **Ramesh Udayashankar Karunagaran** [1,2] and **Dusan Losic** [1,2,*]

1    School of Chemical Engineering and Advanced Materials, The University of Adelaide, Adelaide, SA 5005, Australia; farzaneh.farivar@adelaide.edu.au (F.F.); peilay.yap@adelaide.edu.au (P.L.Y.); ramesh.karunagaran@adelaide.edu.au (R.U.K.)

2    ARC Hub for Graphene Enabled Industry Transformation, The University of Adelaide, Adelaide, SA 5005, Australia

\*    Correspondence: dusan.losic@adelaide.edu.au

†    These authors contributed equally to this work.

**Abstract:** Thermogravimetric analysis (TGA) has been recognized as a simple and reliable analytical tool for characterization of industrially manufactured graphene powders. Thermal properties of graphene are dependent on many parameters such as particle size, number of layers, defects and presence of oxygen groups to improve the reliability of this method for quality control of graphene materials, therefore it is important to explore the influence of these parameters. This paper presents a comprehensive TGA study to determine the influence of different particle size of the three key materials including graphene, graphene oxide and graphite on their thermal parameters such as carbon decomposition range and its temperature of maximum mass change rate ($T_{max}$). Results showed that $T_{max}$ values derived from the TGA-DTG carbon combustion peaks of these materials increasing from GO (558–616 °C), to graphene (659–713 °C) and followed by graphite (841–949 °C) The $T_{max}$ values derived from their respective DTG carbon combustion peaks increased as their particle size increased (28.6–120.2 µm for GO, 7.6–73.4 for graphene and 24.2–148.8 µm for graphite). The linear relationship between the $T_{max}$ values and the particle size of graphene and their key impurities (graphite and GO) confirmed in this study endows the use of TGA technique with more confidence to evaluate bulk graphene-related materials (GRMs) at low-cost, rapid, reliable and simple diagnostic tool for improved quality control of industrially manufactured GRMs including detection of "fake" graphene.

**Keywords:** graphene; graphene oxide; graphite; thermogravimetric analysis





## 1. Introduction

Graphene, a two-dimensional (2D) carbon material with a single layer of sp² carbon arranged in a hexagonal lattice, has been described as the material of 21st century and a new disruptive technology as a result of the combination of its unique properties since its discovery in 2004 [1]. A wide range of graphene applications have been demonstrated in recent years ranging from lightweight composites, functional coatings, adsorbents, electronics, energy storage and etc. [2–5]. Translations of these applications from academia to industry is rapidly increasing in recent years, underpinning fast growing and new emerging graphene industry that apply graphene across many sectors. Manufacturing and supplying high quality graphene materials with known properties for these industrial applications is one of the key obstacles for the growth of this industry. In recent years, many graphene manufacturing processes have been developed and hundred companies worldwide have been established for the industrial production of graphene materials in the forms of powders, pastes or dispersions, which are now available in the market [5,6]. Unfortunately, recent comprehensive quality evaluation studies revealed unexpected findings

that a large percentage of these industrially-produced graphene materials in the market are not as per declared as single or few layer graphene, but rather a mixture of graphene, graphitic and other carbonaceous materials [7–9]. The properties and the quality of these graphene materials are far from the properties of single or few layered graphene defined by the International Organization for Standardization (ISO) [10–12]. These surprising results triggered considerable concerns for down-stream graphene end-users with a potential grave impact on the future of and the growing of multi- million dollars graphene industry and its associated industries, which intend to use this outstanding material. The main concerns are the presence of nanographite (>10 graphene layers) or unexfoliated graphitic particles, highly defective graphene, graphene oxide (GO) and impurities from the production process, which could be overlooked by currently used characterization techniques that focus on single particle analysis. For these reasons, the standardization of graphene and reliable quality control for industrially- produced graphene materials, has now become one of the most critical prerequisites for the growing graphene industry, where the fight against "fake graphene" in the market is one of the top priorities.

Most of the recommended advanced characterization techniques within the current ISO graphene measurement standards being developed, including SEM, TEM, AFM and Raman spectroscopy are localized characterization methods, which are able to probe the properties of graphene materials only for a very small area, such as single graphene particles [10,11,13–15]. These methods are more appropriate for chemical vapour deposition (CVD) graphene layers, but are limited for characterization of graphene powders, which are lacking the representative measure of their key properties at larger bulk quantities. These techniques also require skilled technical personnel, can be time-consuming and not affordable for typical graphene manufacturers, which are usually small start-up companies. Thus, quality control of produced graphene powders is typically performed externally by academic institutions using a limited amount of samples (μg) and localised testing of very few selected graphene particles, which are then used to make the claims of the properties of the product produced at large scale. This level of testing is unable to provide an understanding of the properties of graphene powders when produced in industrial-scale batches (kg or tonne). Ultimately, this leads to a supply chain that lose the credibility on the properties claimed by the graphene manufacturers.

Hence, there is a significant demand to develop and implement a more representable, reliable, simple, and low-cost analytical methods that provides information of bulk properties for industrially produced graphene materials. Several analytical methods such as thermogravimetric analysis (TGA), X-ray diffraction (XRD), surface area (Brunauer–Emmett–Teller, BET nitrogen adsorption/desorption method), particle size distribution (PSD), Fourier-transform infrared (FTIR) spectroscopy and pH titration can meet these requirements [11,13–15]. They could provide valuable information about the properties of graphene powders such as the crystal and graphitic structure, surface area, chemical composition, impurities and functional groups. Among them, TGA, a routinely used analytical technique in the industry for characterization of the thermal properties and impurities of minerals, polymers and carbon materials, is very promising, but has not been explored for the quality control of graphene materials [16–19].

In our recent study, we demonstrate how TGA method can be used as a valuable quality control and analytical tool for the qualitative and quantitative analysis of manufactured raw graphene powders and be used as a simple and reliable method to combat with the "fake graphene" [20]. This study revealed the influence of structural and chemical composition properties of graphene materials such as number of layers, particle size, defects and level of oxygen on TGA/DTG parameters which requires more clarification to make TGA a more reliable method for qualitative and quantitative analysis of graphene materials that can be used as a simple method for the identification of fake graphene. One of the key objectives in the present work is to explore and establish the correlation between key TGA/DTG features and the particle size of GRMs (summarized in Scheme 1), which has not been previously reported. Herein, we systematically investigate the effect of

the particle size of selected GRMs including graphite (Gft), graphene (Gr) and graphene oxide (GO) on their TGA and DTG properties with focus on the thermal decomposition of carbon including their DTG peak range and $T_{max}$ derived from their respective DTG carbon combustion peaks. Three materials including graphite, graphene and GO are prepared and characterized by HRTEM, FESEM, XPS, Raman, XRD and FTIR to determine their key characteristics such as number of layers, defects, chemical composition and crystallinity. Their particle size is determined by two comparative methods including SEM and laser diffraction (LD) techniques, followed by TGA characterization to establish the relationship between the $T_{max}$ values derived from the TGA- DTG peaks of the carbon combustion region with different particle size. Outcomes from this study provide important information about the influence of the particles size of these materials and will help gaining confidence in TGA method for characterization of graphene materials as a quick method for detecting "fake graphene" with large contents of graphitic particles. The established and standardized TGA method will be a valuable, low-cost and simple characterization tool that can be effectively and practically used as a diagnostic tool for industrially manufactured GRMs.

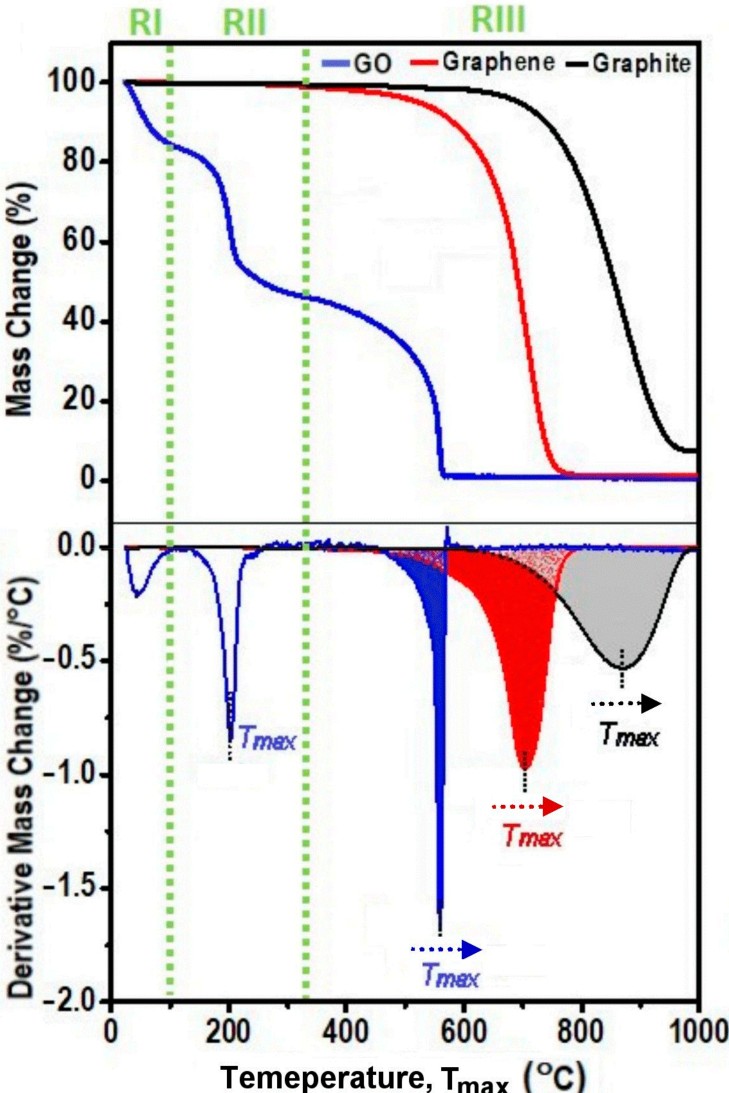

**Scheme 1.** Representative TGA-DTG graphs and the $T_{max}$ values of graphene, GO, and graphite, which are used as foundation to explore the influence of particle size.



## 2. Materials and Methods

### 2.1. Materials and Sample Preparation

Graphite used in this study was in the form of graphite flakes provided by a local graphite mine company (Uley, Eyre Peninsula, SA, Australia). Prior to TGA measurement, graphite flakes were crushed and sieved to different flake sizes including ≤25 μm, 25–53 μm, 53–100 μm and 100–150 μm. Commercial graphene materials provided by undisclosed company with different particle size (7 to 74 μm) were processed using tip sonication technique [21]. For the preparation of GO samples, non-sieved graphite powder, powder sieved at ≤25 μm and 25–53 μm were reacted with oxidising agents to form GO through improved Hummer's method according to our published work previously [22]. More details about the samples are tabulated in Table S1. To minimize the discrepancy arise from other factors such as oxygen level, morphology, level of defects and thickness, the three main carbon materials studied in this work were processed from their corresponding parent material using the same preparation method. For instance, all the GO prepared using the same graphite flakes which were sieved to different size before undergoing the same oxidation and exfoliation process.

### 2.2. Characterization Techniques

2.2.1. Thermogravimetric Analysis

TGA analyses of prepared samples (~5 mg) were performed using a Mettler Toledo TGA/DSC 2 with heating rate at 10 °C/min in air condition at flow rate of 60 mL/min. Collected TGA data demonstrated typical mass loss with regards to temperature and their corresponding first derivative (DTG) graphs provided information on mass loss rate versus temperature. TGA-DTG plots have been popularly applied to determine the composition of water, volatile compounds, combustible carbon and impurities in graphene and its related materials. In this study, we focused on the thermal parameter (temperature of maximum mass change rate, $T_{max}$) by fitting the TGA-DTG curves using Origin software. $T_{max}$ of the final mass loss step related to the carbon combustion can be determined from the maximum peak position in the negative region of the DTG curve (Scheme 1).

2.2.2. Scanning and Transmission Electron Microscopy (SEM and TEM)

Lateral size and morphology of the representative carbon materials were determined using scanning electron microscope (FE-SEM, Quanta 450 FEG, FEI, USA; and SEM, Hitachi SU1510, Japan) at an operating voltage of 10–30 kV and a transmission electron microscope (TEM) at 120 kV (TEM, FEI Tecnai G2 Spirit, FEI, USA; Philips CM200, Japan at 200 kV). For SEM imaging, samples were dispersed in ethanol (GO and Gft) and isopropanol (Gr) by gentle sonication and drop casted on a carbon tape, followed by coating the sample with *ca* 5 nm of Pt coating. For TEM imaging, the dispersed samples were drop-casted on a lacey carbon film on Cu grid. For SEM, at least images of 50 particles were obtained (at 3 different spots) and at for TEM least 20 particles (at 3 different spots) for each of the tested sample. The lateral size (LxW) of each particle was measured from obtained images with the average of the two values was presented for their average lateral size. One image is selected to show the representative particle size for each material.

2.2.3. Raman Spectroscopy

Raman spectrometer (LabRAM HR Evolution, Horiba Jvon Yvon Technology, Japan) with a 532 nm laser (mpc3000) was used to understand the vibrational characteristics and distinguish the carbon materials tested in this work. The Raman spectra were collected at 500 to 3000 cm$^{-1}$ with an integration time of 10 s for three accumulations using a $100\times$ objective lens and laser spot was 721.16 nm. Raman spectra were obtained from at least from 5 different particles and their 3 spots (center and edge). One representative Raman graph is presented.

### 2.2.4. FTIR Spectroscopy

Functional groups present in the carbon samples were confirmed using FTIR spectrometer (Nicolet 6700, Thermo Fisher) with the spectra collected in the range of 500–4000 cm$^{-1}$.

### 2.2.5. Powder X-ray Diffraction (XRD)

Powder X-ray diffractometer (600 Miniflex, Rigaku, Japan) equipped with a Cu X-ray tube ($\lambda$ = 1.54 Å, 40 kV and 15 mA) was run at with 10° min$^{-1}$ scan speed in the range of 2θ = 5 to 80° to determine the interlayer spacing of the examined carbon materials.

### 2.2.6. X-ray Photoelectron Spectroscopy (XPS)

X-ray Photoelectron Spectroscopy (XPS, AXIS Ultra DLD, Kratos, UK) equipped with a monochromatic Al Kα radiation source (hv = 1486.7 eV) was applied to identify the presence of impurities and the chemical composition of the materials at 225 W, 15 kV and 15 mA. XPS survey spectra were measured at 0.5 eV step size over −10 to 1100 eV at 160 eV pass energy with peak fitting analysis performed using Casa XPS$^{TM}$ software. The core-level XPS spectra were calibrated to the adventitious carbon at 284.8 eV.

### 2.2.7. Particle Size Distribution (PSD) by Laser Diffraction (LD)

Particle size distribution (PSD) of the samples was determined using the principle of laser light diffraction (LD) on a Mastersizer 2000 (Malvern Instruments, Malvern, UK) with refractive index and absorption set at 2.42 and 1, respectively, using water as the dispersant. Sample (fine powder) was gradually added into a tank containing water until obscuration (~5%) was achieved and all the measurements were carried out without ultrasonication to preserve the primitive size of the carbonaceous particles. PSD graphs were plotted based upon the median particle size by volume, d(50)$_v$, with three measurements collected for each sample.

## 3. Results and Discussion

### 3.1. Structural and Chemical Characterization of GO, Graphene and Graphite

The lateral size and morphology of prepared GO, graphene and graphite materials with different particle size used in this TGA study were verified by combination of FESEM and PSD/LD characterizations. FESEM analysis was used in this work to reflect the actual particle size and shape of GRMs with limitations of selected number of imaged particles, which do not represent whole bulk powder samples. For that reason, complementary PSD/LD method was applied to measure the particle size in dispersion despite that this method assumes the measured 2D sheets as spherical particles.

Typical FESEM images of selected single particle to represent the average particle size of GO, graphene and graphite samples are summarized in Figure 1, Figure S1 and Table 1. FESEM images of GO exhibited highly wrinkled GO sheets with their lateral size in two lateral dimensions recorded (Table 1) prepared after extensive oxidation and exfoliation process. Representative image of graphene samples with different particle size captured by SEM illustrated the isolated graphene sheets with varied lateral size obtained. The thickness of prepared graphene layers was confirmed in the range of 3–7 layers (Figure S2), suggesting <10 layers of Few Layer Graphene (FLG) grade was used throughout this work. Typical SEM images of individual graphite particles sieved with different sieves fractions were presented in Figure 1k–n.

Comparative particle size measurements of these samples using LD method with their PSD curves are also presented in Figure 1d,j,o. Note that the particle size was expressed in median particle size by volume, d(50)$_v$, where the particle size by volume distribution can be defined as the relative proportion of volume occupied by particles with different size denoted by the peak of the curve [23,24]. The tested GO, Gr and Gft samples demonstrated a monodisperse population particle pattern with d(50)$_v$ recorded at 28–121 μm for GO, 7–74 μm for Gr and 24–150 μm for graphite, respectively. In this study, the measured particle size from SEM technique was found to be in good agreement with the

measurements from LD method that validates the applicability of LD technique to probe the particle size of GRMs although there is a controversy on the reliability and accuracy of the particle size measurement arise from indirect particle sizing techniques including dynamic light scattering (DLS) and LD for GRMs [24,25]. For instance, a well-defined correlation between TEM-measured nanosheet length and DLS-measured hydrodynamic radius to estimate the lateral size of 2D nanosheets (graphene, $MoS_2$ and $WS_2$) was reported by M. Lotya et al. [26]. Nevertheless, DLS method may not be accurately applied to determine the nanosheet size in view of its high relative error (40%) in this study [26]. In contrast, LD method was found to offer high accuracy and comparable PSD data with SEM technique for GO particles with only maximum deviation of 7–9% recorded despite that this method is claimed to be limited for spherical particles and underestimates the presence of particles with small size [25]. Similar to our work, this recent study strongly supports the applicability of LD technique to provide the PSD information of bulk GRMs considering the low cost, simple, rapid and high throughput of LD technique. For the purpose of this study, PSD from LD technique is mainly applied to demonstrate the trend of the particle size of GRMs apart from exploring the dependence of their particle size on the TGA-DTG features despite that it is not a commonly used technique to determine the particle size of GRMs. The particle size of the GRMs studied in this work is compared between direct (SEM) and indirect (LD) particle sizing techniques (Table 1, Figure 1 and Figure S1). Note that the PSD curves produced using LD technique are highly reproducible with nearly zero standard deviation recorded for all the three repeated LD measurements as depicted in Figure S3. This strongly suggests the high reproducibility of applying this particle sizing technique to determine the particle size of GRMs.

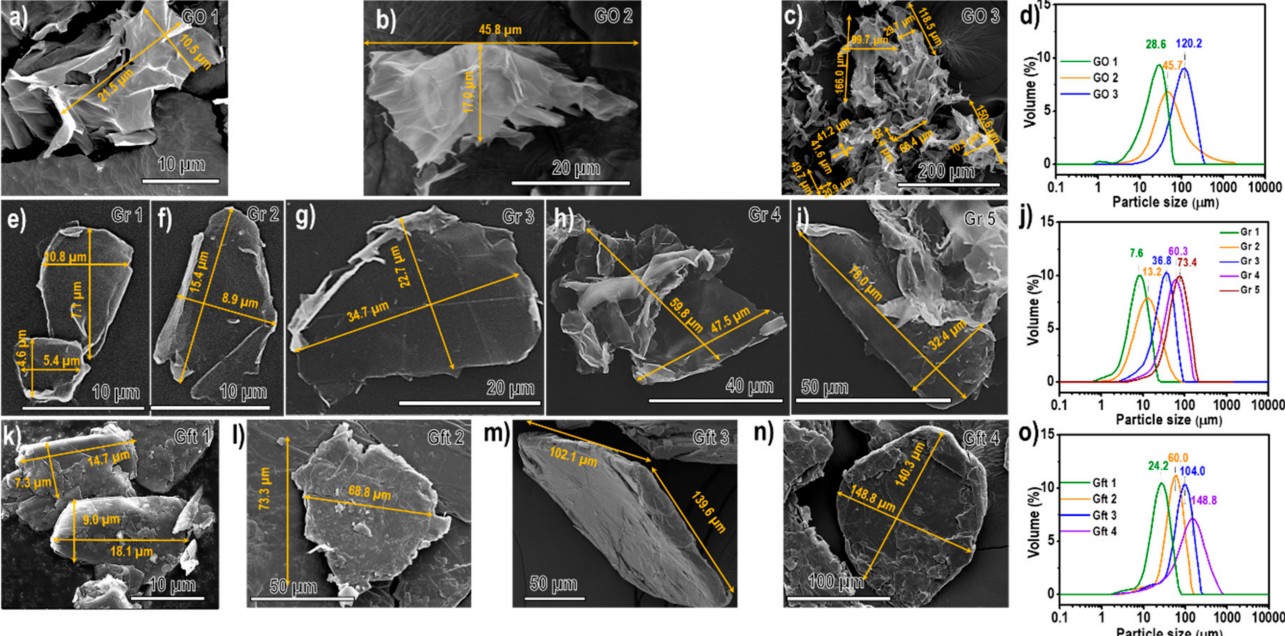

**Figure 1.** Representative FESEM images with PSD/LD plot of (**a–d**) GO 1–3, (**e–j**) Gr 1–5 and (**k–o**) Gft 1–4. Direct imaging technique (FESEM) was adopted to represent the size of GRM particles based on the 2D lateral dimensions as indicated by the orange solid line; Typical LD-particle size distribution (PSD/LD) plots based on median particle size by volume, $d(50)_v$ where the peak of the curve determined from their graphene, GO and graphite dispersions.

Figure 2a showed the characteristic Raman peaks of GO, graphene and graphite. The ratio of the peak intensity of the D to G bands ($I_D/I_G$) at around 1350 $cm^{-1}$ and 1580 $cm^{-1}$, respectively, indicates the level of disorder or defect present in the carbon materials (Figure 2a) [28]. As expected, all the GO samples in these studies are highly defective with $I_D/I_G$ ratio recorded between 0.87–0.91, indicating that these samples have

undertaken rigorous exfoliation and oxidation process. Meanwhile, graphene samples exhibited $I_D/I_G$ ratio of 0.19–0.33, suggesting that these graphene materials experienced mild level of defects compared to graphite that could be resulting from the exfoliation process [29]. Of all the carbon materials, graphite samples appeared to be the least defective materials with $I_D/I_G$ ratio of <0.1, which could be explained by less materials processing stages involved except for physical sieving process step. Our Raman spectroscopy findings are in good agreement with the thickness outcome determined by HRTEM analysis (Figure S2), confirming that the graphene studied in this work is FLG as evidenced by a distinguished and symmetric 2D band at about 2700 cm$^{-1}$ detected for the graphene sample; while a broader and asymmetric 2D peak alongside with a shoulder peak observed at around 2720 cm$^{-1}$ for graphite sample [30].

**Table 1.** Measurement results for comparative PSD/LD and SEM particle size characterization of the series of GO, graphene and graphite used in this study. Average lateral size measurements were performed following ISO/TS 21356-1:2021 [27] using in ate least 20–50 particles taken from SEM and TEM images at least 3 different spots. Results from PSD/LD measurements are presented as an average plus stdev are based on 3 LD measurements.

| | GO | | | Graphene | | | | | Graphite | | | |
|---|---|---|---|---|---|---|---|---|---|---|---|---|
| | GO1 | GO2 | GO3 * | Gr1 | Gr2 | Gr3 | Gr4 | Gr5 * | Gft1 | Gft2 | Gft3 | Gft4 * |
| Sieve fraction/μm | <25 | 25–53 | un-sieved | | | N.A. | | | <25 | 25–53 | 53–100 | 100–150 |
| Average lateral size, SEM ± stdev (μm) | 23.5 ± 10.6 | 46.4 ± 21.2 | 73.4 ± 40.4 | 7.0 ± 2.8 | 15.9 ± 3.7 | 28.3 ± 0.5 | 49.4 ± 6.0 | 52.8 ± 3.4 | 11.3 ± 2.1 | 44.9 ± 37.0 | 130.6 ± 13.7 | 158.8 ± 20.2 |
| Average d(50) from LD ± stdev (μm) | 28.6 ± 0.010 | 45.7 ± 0.077 | 120.2 ± 0.032 | 7.6 ± 0.028 | 13.2 ± 0.018 | 36.8 ± 0.043 | 60.3 ± 0.060 | 73.4 ± 0.100 | 24.2 ± 0.099 | 60.0 ± 0.028 | 104.0 ± 0.056 | 148.8 ± 0.006 |

Structural properties of the carbon materials were studied using powder XRD technique as presented in Figure 2b. Pristine graphite sample showed a strong and well-defined peak appeared at $2\theta = 26.6°$, while this peak also found at the same position but with a weaker and lower intensity in graphene parent sample. Upon the intercalation of oxygen groups using strong oxidizing agents, a massive peak shift was detected from $2\theta = 26.6°$ to $10.8°$ in GO samples, confirming an increased interlayer spacing between the graphene layers due to insertion of oxygen functional groups such as epoxide, hydroxyl and carboxylic groups [22,31]. Similarly, all their respective counterpart materials exhibited the same XRD pattern (Figure S4).

FTIR analysis of the carbon materials (Figure 2c) further corroborated the XRD results with the characteristic transmittance bands for oxygen functional groups including C=O (1720 cm$^{-1}$), -OH (3000–3500 cm$^{-1}$) and C-O-C (1228 cm$^{-1}$ and 1048 cm$^{-1}$) found in GO. Nearly a flat line with no obvious peak was detected for both Gft and Gr, suggesting no significant amount of oxygen functional groups was found in these two carbon materials [22,32].

XPS analysis further substantiated the discussed characterization outcomes with the elemental quantification performed on the parent graphene and its associated materials determined from their survey scans. Based on the curve fitting analysis, relatively high carbon content can be observed for Gft (94.2 atomic %) and Gr (97.7 atomic %) compared to GO (68.1 atomic % C). As expected, GO has the highest oxygen content (31.9 atomic % O), followed by Gft (5.2 atomic % O) and Gr (2.4 atomic % O).

*3.2. Elucidating the Impact of Particle Size of GO, Graphene and Graphite on Their TGA Characteristics*

Prepared GO, graphene (Gr) and graphite (Gft) samples with different particle size were subjected to thermal decomposition up to 1000 °C in the air atmosphere to probe their particle size dependence thermal characteristics. Representative TGA graphs are

presented in Figure 3, showing TGA graphs on the top and their corresponding first derivative thermogravimetry DTG graphs on the bottom with their respective $T_{max}$ values. An example demonstrating the plot of TGA-DTG curves with the derivation of thermal parameters can be found in Figure S5. These graphs representing the thermal pattern of GO, Gr and Gft clearly showed that all these materials were completely burnt in the air atmosphere before 1000 °C, leaving no residue behind that suggests their high purity. Subsequent DTG graphs generated from TGA curves demonstrated three distinguished DTG peaks corresponding to three mass loss events (GO) as well as a typical DTG peak that can be related to only a single mass loss step occurred during the thermal degradation for both Gr and Gft.

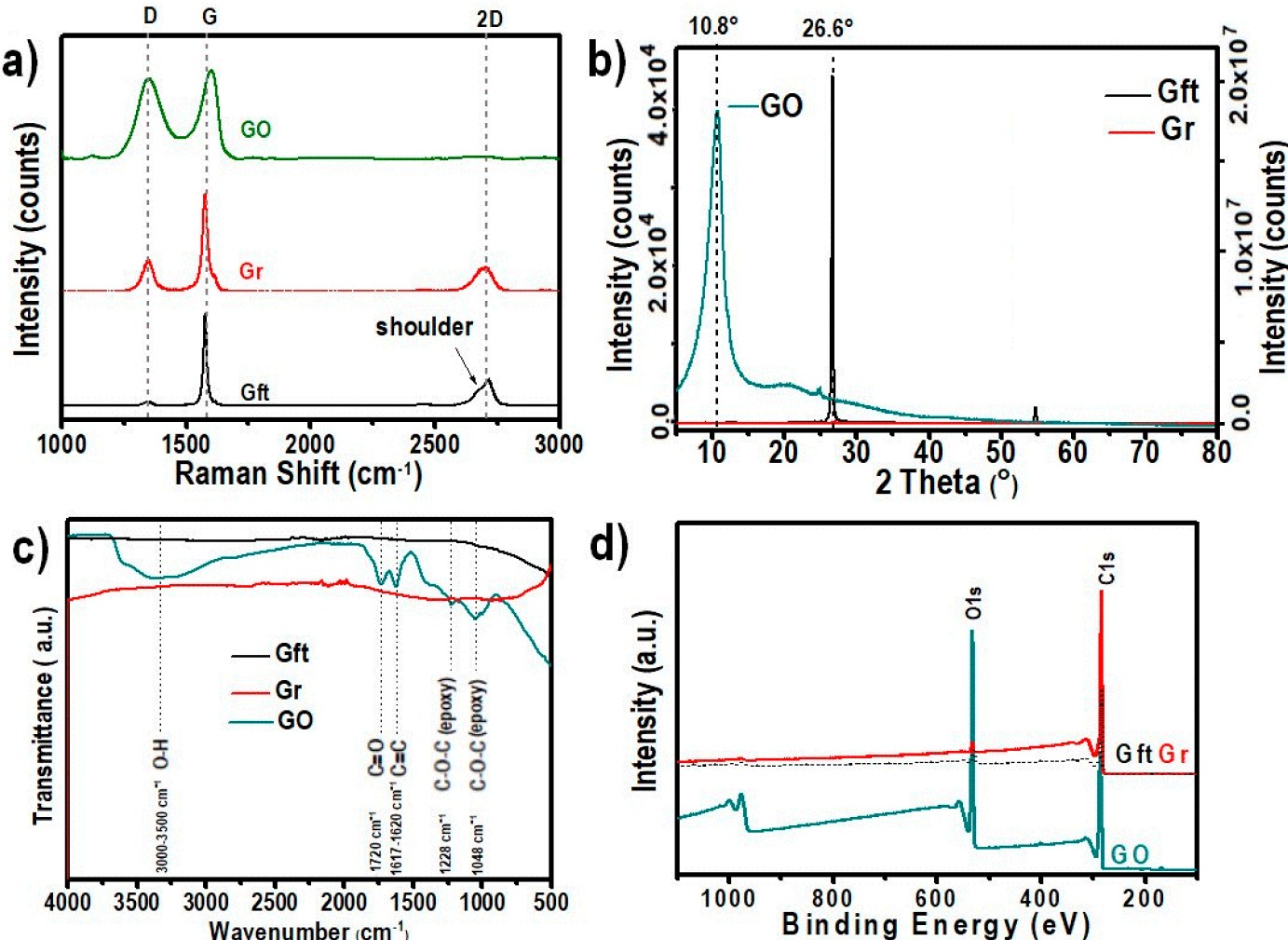

**Figure 2.** (**a**) Raman spectra, (**b**) XRD diffractograms with a right inset showing comparative intensity of Gr and Gft, (**c**) FTIR spectra and (**d**) XPS survey plots with an inset of atomic composition of parent GO, Gr and Gft materials.

The key mass loss events of GO can be explained based on their decomposition temperatures: <100 °C for water elimination, 100–360 °C due to the removal of oxygen functional groups and 360–1000 °C can be linked to the oxidative pyrolysis of carbon framework [33]. It is worth mentioning that the first and second DTG peaks, representing the loss of water and oxygen functional groups events, did not vary significantly with different sieved size of GO samples. Similar to the third mass loss step of GO, the only DTG peak found in graphene and graphite samples can be ascribed to the combustion of carbon in the air condition. These DTG peaks appeared to change noticeably when varied size of carbonaceous materials were analyzed, which is the focus of our present study that

will further unveil in the next sections. As depicted in the DTG graphs (bottom row of Figure 3a–c), their carbon combustion temperature was distinctly marked in terms of the temperature of maximum mass change rate ($T_{max}$) with GO experienced the earliest carbon combustion ($T_{max}$ = 550–616 °C), followed by graphene ($T_{max}$ = 650–713 °C) and graphite ($T_{max}$ = 840–950 °C). The distinguished $T_{max}$ can be attributed to the maximum of the external heat energy required to overcome the strong bonding within their carbon lattice structure. The observed lower $T_{max}$ in GO indicated that less heat energy is required to overcome weaker non-graphitic $sp^3$ hybridized carbon with a high density of defects after the rigorous oxidation reaction compared with graphene and graphite. GO also has high level of oxygen groups, which is an additional reason for lower level of energy for their decomposition. Graphene has higher $T_{max}$ because it needs larger amount of heat energy to break down the $sp^2$ hybridized carbon atoms ordered by covalent bonds in a hexagonal carbon framework. Graphite, the most thermodynamically stable carbon materials studied in this work, demands even more heat energy due to its strongest 3D carbon network, consisting of a large number of graphene stacked layers held by additional van der Waals forces [21]. From these studies, it is obvious that $T_{max}$ can be regarded as a key parameter that can be used for their identification and specifically for detection of presence of GO and graphite impurities in graphene materials as summarized in Table S2.

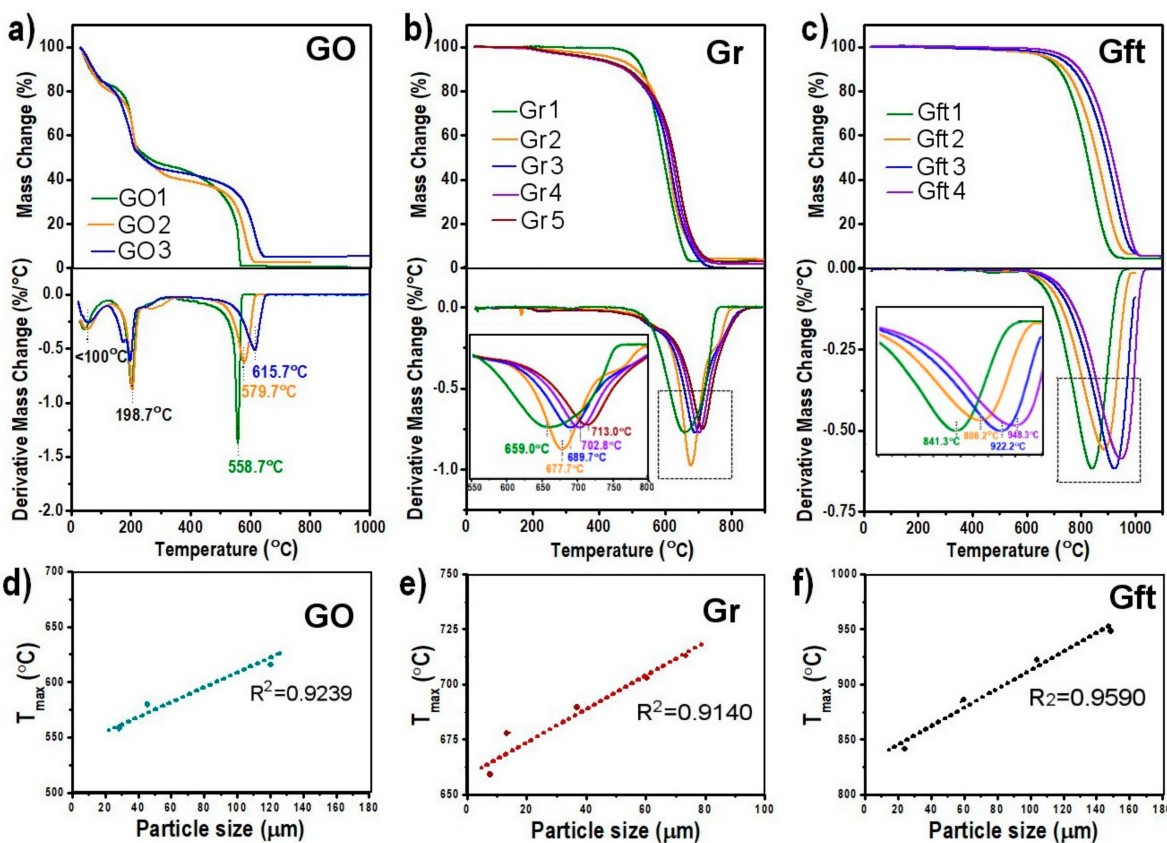

**Figure 3.** Plots of TGA (top row)-DTG (bottom row) for an array of carbonaceous materials with varied sizes: (**a**) GO, (**b**) grapheme * and (**c**) graphite * samples, showing the $T_{max}$ values derived from their respective DTG peaks in the carbon combustion region. * Inset illustrates closed-up image of rising $T_{max}$ values as particle size of GRMs increased. Plots of $T_{max}$ values vs the median particle size, d(50)$_V$, determined by PSD/LD technique of (**d**) GO, (**e**) Gr and (**f**) Gft samples.

The obtained $T_{max}$ values of GO, graphene and graphite were plotted as a function of median particle size obtained from PSD/LD measurement and presented in Figure 3d–f. As explicated in the previous section, the substantial rising $T_{max}$ trend visualized on the three types of tested carbonaceous materials with GO series exhibited the lowest $T_{max}$ range (550–616 °C), followed by 650–713 °C for Gr samples and Gft batches recorded the

highest $T_{max}$ range (840–950 °C) can be linked to their respective carbon building blocks involved in the bonding and structural framework. Regarding the influence of particle size, it was discovered that with the increase of the particle size, the $T_{max}$ representing the carbon combustion region shifted to the right, resulting a higher $T_{max}$ values for all the characterized materials. Taking GO as example, as the particle size increases from 28.6 μm, to 45.7 μm and 120.2 μm in GO 1–3, the $T_{max}$ values in the DTG plot of its carbon combustion region (third mass loss step) in GO gradually shifted from 558.7 °C to 579.7 °C to 615.7 °C (Figure 3d). In the case of Gr samples, PSD analysis showed that the measured particle size 7.6 μm, 13.2 μm, 36.8 μm, 60.3 μm and 73.4 μm can be well-correlated to their growing $T_{max}$ values, 659.0 °C, 677.7 °C, 689.7 °C, 702.8 °C and 713.0 °C, respectively (Figure 3e). Similarly, as the particle size of tested graphite samples increased from 24.2 μm, 60.0 μm, 104.0 μm to 148.8 μm, an increasing of $T_{max}$ values of 841.3 °C, 886.2 °C, 922.2 °C and 948.3 °C trend can be observed as evidenced in Figure 3f. A key message from this study is that there is a significant difference in the $T_{max}$ value between graphene and its key impurities including GO and graphite and their particle size do not overlap with the $T_{max}$ value of graphene as presented in Figure 4. Hence, there is no potential error resulting from this TGA-DTG method to identify these impurities in industrial graphene materials regardless of their particle size.

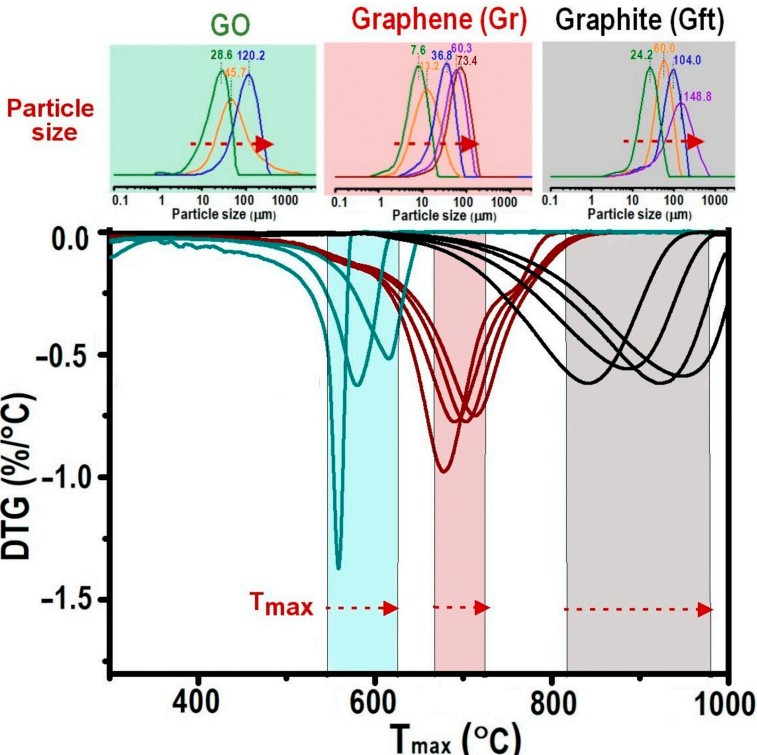

**Figure 4.** Summary of TGA-DTG graphs for GO, graphene and graphite with different particle size showing non-overlapped $T_{max}$ values and the ability of TGA-DTG method to distinguish GO and graphitic impurities from graphene for their quality control.

The correlation analysis performed on $T_{max}$ vs particle size graphs (Figure 3d–f) showed that high coefficients of determination (adjusted $R^2$ values > 0.9) based on the $T_{max}$ values in all the three samples as tabulated in Table S3. Regardless of the type of carbon materials, this finding strongly suggests that the $T_{max}$ values derived from the DTG peak in the carbon combustion region can be linearly correlated to the particle size of the carbon materials studied. The observed steadily increasing linear trend can be explained by the large surface area of these materials that needs more heat energy for their thermal decomposition. As supported by W. Jiang et al. and M. Shtein et al., carbonaceous

particles with smaller size possess higher amount of edge sites per mass unit, this means that smaller carbon particles are more reactive and susceptible towards combustion process compared to particles with larger size [14,19]. This also implies that carbon materials with larger particle size tend to shift their $T_{max}$ to higher temperature because of slower combustion kinetics in air, which is also evidenced in our studies [19]. Our comprehensive evaluation over different types of carbonaceous materials clearly showed that particle size of carbonaceous materials can be linearly correlated to the $T_{max}$ values derived from their corresponding DTG peaks in the carbon combustion region, which is in good agreement with the reported literature [14,19].

## 4. Conclusions

In summary, in this study, we presented the influence of particle size of GO, graphene and graphite with the aim to establish correlation of key TGA-DTG thermal parameter ($T_{max}$) that can be used for qualitative determination of bulk graphene powders and their impurities such as graphite and GO particles. The $T_{max}$ values derived from the DTG peak of the carbon combustion region exhibited a direct and linear relationship with the particle size of the carbonaceous materials regardless of the type of carbonaceous materials (graphene, GO and graphite) as confirmed by their respective high R-squared values ($R^2 > 0.91$) achieved in their linear regression analyses. A significant difference in the $T_{max}$ value can be identified between graphene and its key impurities (GO and graphite) and their particle size, where their $T_{max}$ values do not overlap with the $T_{max}$ value of graphene that rules out the possibility to have erroneous results due to deviation from particle size to identify these impurities in industrial graphene materials using the TGA-DTG approach. This study shows the potential of TGA analysis to be used as a low-cost screening tool for the quality control and characterization of manufactured graphene materials and their products at an industrial scale, which has been overlooked previously.

**Supplementary Materials:** The following are available online at https://www.mdpi.com/article/10.3390/c7020041/s1.

**Author Contributions:** Conceptualization, D.L., F.F. and P.L.Y.; Methodology, F.F. and P.L.Y.; Formal analysis, F.F., P.L.Y. and R.U.K.; Investigation, F.F. and P.L.Y.; Writing—Original Draft Preparation, F.F., P.L.Y. and D.L.; Writing—Review & Editing, D.L. and P.L.Y.; Resources, D.L.; Supervision, D.L.; Project Administration, D.L.; Funding acquisition, D.L. All authors have read and agreed to the published version of the manuscript.

**Funding:** This research was funded by the ARC Research Hub for Graphene Enabled Industry Transformation (IH150100003).

**Institutional Review Board Statement:** Not applicable.

**Data Availability Statement:** All obtained Data are available at The University of Adelaide Data Storage system.

**Acknowledgments:** The authors thank the Australian Microscopy and Microanalysis Research Facility (AMMRF) for the facilities access and technical support of TEM (Ashley Slattery); and XPS (Chris Bassell) at the Microscopy Australia Facilities (NCRIS scheme) situated at the University of South Australia.

**Conflicts of Interest:** The authors declare no conflict of interests.

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
