# Peer review of "Thermogravimetric Analysis (TGA) of Graphene Materials: Effect of Particle Size of Graphene, Graphene Oxide and Graphite on Thermal Parameters"

_carbon_

Round 1
Reviewer 1 Report
Overall, I felt this paper had merit and could be an important contribution to the field. I did have issues with some of the conclusions drawn because I felt there was a lack of data to support those conclusions. I hope the following comments can be used to improve the manuscript.
- Abstract - Define GRMs.
- Abstract - You reference an increase in material combustion temperatures, but then put the temperatures in reverse order then add particle size as part of the mix. I would clarify this by breaking it down to two sentences - one that emphasizes the change in oxidation temperature range and the other that discusses the relationship between particle size and temperature.
- Introduction, sentence 1 - This sentence does not make sense.
- Introduction "underpinning fast growing of new emerging graphene industry" - English needs to be improved.
- Introduction, use of "alarming" - I don't find the lack of well-produced materials alarming. I would suggest a different word here.
- Title vs text - you use Thermogravimetric in the text versus Thermal Gravimetric in the title. Pick one and be consistent.
- Pg 2, intro "the rationale was based on the confirmed fact that key..." This is a bold statement. I don't think you provide evidence for this and it should be removed.
- Page 3 - Define GRMs. Many typos, English grammar issues in this paragraph.
- Section 3.1 - I do not believe your conclusions/results in this section. You seem to image an inadequate number of particles to represent the population (2-3 samples???). I would like to see far more particles analyzed, statistical data as to how well the particle sizing and image analysis match up and more conclusive results from TEM work to demonstrate the particle layers.
- I would like to know more details about the standard deviation on Figure S3,l. Is that significant?
- Sentence "As tabulated in Table 1..." This is repetitive and does not add to the text.
- What does "nearly no standard deviation" mean to you? I would prefer numbers.
- Many of the figures and tables are unreadable because the text is too small it is unreadable.
- Table 1 - Am I reading correctly that many of these only have 2-3 repeats? This is inadequate for particle counting.
- How many TEM samples were done? Is it just the one?
- For the XRD results, please include the not shown results in SI.
- From the TG, it seems there is a lot of water. This is not my experience with these materials. What do you think caused this? Why is there so much?
- The listed PSD numbers can be presented in a better fashion on page 9. Add these numbers into a table rather than listing.
Author Response
Attached in response letter

Reviewer 2 Report
Farivar and et al. proposed and experimentally demonstrated a relatively simple method for the qualitative characterization of industrial powders of graphene, graphene oxide (GO), and graphite. The temperature ranges of decomposition of graphene, GO, and graphite were determined by thermogravimetric analysis. The Tmax values ​​obtained from the thermogravimetric analysis of the carbon combustion region show a direct and linear dependence on the particle size of carbonaceous materials regardless of the type of carbonaceous materials (graphene, GO and graphite). It has been experimentally shown that, regardless of the particle size, the Tmax values ​​obtained for graphene, graphene oxide, and graphite do not overlap.
In my opinion, the article can be published in the C.
However, the manuscript requires revision in accordance with the following notes:
- It is stated that in the studies of Raman spectra, a 100 × objective was used, while the size of the investigated spot was 100 µm. This requires clarification. Typically, in Raman spectrometers, when using a 100 × objective, the size of the focused laser beam is less than a few micrometers.
- The X-ray wavelength of the diffractometer used needs to be specified.
- Presented in fig. 2 (a) Raman spectrum is not typical for graphene, consisting of SEVERAL LAYERS for the following reasons: 2D peak is too wide, practically does not differ from the width of 2D peak of graphite; the ratio of the 2D intensity to the G peak intensity is small.
- It is not clear to me why there is no peak at the angle 2? = 26.6 deg in the diffractogram (see Fig. 2 (b)) for graphene. The authors claim that it is there, but it has low intensity. Why was it impossible to take a diffractogram with so much graphene that this peak was large enough?
- In Fig. 2 (a) - 2 (d) there are no tick marks on the ordinate axes. Therefore, the notation (a.u.) is meaningless.
- The insets shown in fig. 3 (b) and 3 (c), as well as the upper part of Fig. 4 are too small. Tick labels in these figures are not legible. Therefore, they need to be redone.

Author Response
Attached in response letter

Round 2
Reviewer 1 Report
After seeing the modifications made, I think some items were improved from the previous review, but there are still a lot of uncertainties regarding the particle size. Since this is a major factor they are basing their particle ranges on, I think it requires further examination. I still find Table 1 to be more confusing than useful and I think the authors should reconsider their presentation of the data they have collected. In addition, all Supplementary Information data should have error bars, especially in the case of particle measurements.
Author Response
Reviewer#1 comments
After seeing the modifications made, I think some items were improved from the previous review, but there are still a lot of uncertainties regarding the particle size. Since this is a major factor they are basing their particle ranges on, I think it requires further examination. I still find Table 1 to be more confusing than useful and I think the authors should reconsider their presentation of the data they have collected. In addition, all Supplementary Information data should have error bars, especially in the case of particle measurements.
Authors’ response:
To measure particle size of graphene materials in this paper we used combination of two most common and recommended methods. One is using SEM/TEM imaging of dispersed graphene particles on Si wafer as defined by International standard ISO/TS 21356‑1:2021 where the size of 2 dimensional graphene particles (LxW length X Width) were manually measured from images and presented by one dimension plus st. dev. This is defined and recommended method by International ISO standard to presenting lateral size of graphene particles from images which is maybe confusing for the reviewer.
The second method was using PSD/LD by measuring average particle size of graphene dispersed in solution where average particle size distribution (D50) was obtained by software based on measuring the angular variation of the light small angles relative to the laser beam and small particles scatter light at large angles considering the graphene particles are spherical.
Merits and limitations of both methods are explained in paper and known in literature which are reasonable in good agreement. In this paper we used PSD/LD data to generate graphs to show the influence of particle size on TGA parameters and we are confident that all presented results are correct.
Changes in revision 2:
- The graphs for PDS/LD measurements in 3 replicates for samples are presented in Figure S3 showing negligible differences in their graphs which average values were presented in Table 1 with dev as zero point (only two decimals were considered). The std. dev. values have already been included in our previous revision documents but additional decimal places have been included to address the response from reviewer and included in Table 1 (row 3)
- Calculation of the surface area of particles measured by SEM/TEM method is removed from the Table 1 to avoid confusion.
